# The Design and Impact of a Clinic-Based Community Program on Food Insecurity, Healthy Eating Behaviors, and Mood

**DOI:** 10.3390/nu15204316

**Published:** 2023-10-10

**Authors:** Tiffany Wesley Ardoin, Elizabeth Perry, Chelsea Morgan, Jared Hymowitz, Donald Mercante

**Affiliations:** 1LSU Health Sciences Center Baton Rouge, Department of Internal Medicine, Baton Rouge, LA 70808, USA; 2Our Lady of the Lake Hospital, Geaux Get Healthy, Baton Rouge, LA 70808, USA; 3American Heart Association, Community Impact, Baton Rouge, LA 70808, USA; 4HealthyBR, The Mayor’s Healthy City Initiative for Baton Rouge, Baton Rouge, LA 70802, USA; 5LSU Health Sciences Center New Orleans, School of Public Health, New Orleans, LA 70112, USA

**Keywords:** food insecurity, food security, nutrition security, nutrition education, food access, social determinants of health, community program, food assistance, nutrition resources

## Abstract

Food insecurity is a national issue that disproportionately impacts Louisiana citizens, contributing to the state’s poor health outcomes. We know that the Supplemental Nutrition Assistance Program (SNAP) and food pantries improve access to food, but we have limited data on what interventions improve food insecurity. The Geaux Get Healthy Clinical Program at Our Lady of the Lake (GGHOLOL) is a clinic-based community program that leverages community partnerships and a clinical setting to provide education and access to resources for individuals with food insecurity. This prospective study examines the impact of GGHOLOL on food insecurity as a pre-post survey evaluation over a two-year period. A total of 57 research participants with food insecurity completed the program. Mean food security scores improved at completion of GGHOLOL, and these scores further improved 6 months after enrollment. Furthermore, participants demonstrated sustainable improvements in healthy eating, cooking, and shopping behaviors. Lastly, participants improved their overall depression scores at the completion of the program with sustainable improvement at 6 months. With the improvement in GGHOLOL on food insecurity and nutrition behaviors, GGHOLOL may serve as a model for other programs addressing food insecurity in the future.

## 1. Introduction

Food insecurity, “the inability to afford nutritionally adequate and safe foods” [1], affects a significant portion of the American population on a daily basis. In Louisiana, the prevalence of food insecurity is particularly high, with a rate of 14.5% in 2022, surpassing the national average of 10.4% and ranking behind only Mississippi and Arkansas in terms of prevalence [2]. Certain populations, including Blacks, Hispanics, single individuals, less educated individuals, and those who live in households with children, experience a higher prevalence of food insecurity, with rates as high as 21.7% among black, non-Hispanic households and 27.7% in single women households with children [3,4].

Food insecurity is significantly associated with many health conditions and poor health outcomes. While it is directly linked to adiposity, it is also associated with depression, anxiety, hyperlipidemia, diabetes, and hypertension independent of the body mass index (BMI) [1,3,5]. In a retrospective observational study utilizing NHANES data, households with significant food insecurity had a higher incidence of hypertension (36.1%) compared to those without food insecurity (19.7%) [6]. Moreover, food insecurity is correlated with an increased risk of chronic health conditions such as coronary heart disease, hepatitis, stroke, cancer, asthma, COPD, and chronic kidney disease [6]. These health implications contribute to Louisiana’s low rankings in heart disease mortality (46th), stroke mortality (47th), diabetes prevalence (47th), and obesity prevalence (47th) among the 50 states [7].

Given its impact on health, food insecurity has gained attention from various stakeholders within U.S. Healthcare systems, including hospital systems, payors, and health policymakers [8]. It is a multifaceted issue that intersects with fields such as economics, health policy, primary care, social work, nutrition, and public health. While food insecurity is an ongoing issue throughout the U.S. population, its prevalence becomes more noticeable during periods of economic stress and can vary for individuals based on personal circumstances and other social determinants of health. Addressing food insecurity is complex due to the diverse range of factors contributing to its fluctuating nature, spanning from the production and distribution of food to individual challenges such as living in a food desert, financial constraints, or lack of knowledge and resources when purchasing and preparing nutritious meals [9].

Several interventions have shown success in addressing food insecurity, such as the Freshplace Intervention Model, which incorporates food pantry choices, motivational interviewing, and access to additional services and resources [10]. The Supplemental Nutrition Assistance Program (SNAP) has also been associated with improved food access, although its impact on food quality remains uncertain [11]. However, data on community interventions and their effects on food security are limited. While established food insecurity programs exist within large healthcare systems like ProMedica and Geisinger, published data on this program’s impacts are scarce [12]. A systematic review and meta-analysis published in JAMA in 2021, which examined food insecurity interventions and their associations with food security status and health outcomes, revealed a dearth of evidence utilizing 39 studies of heterogeneous interventions and outcomes [13]. Although there is high-certainty evidence that directly providing food reduces food insecurity, data on other forms of intervention or their combinations, such as providing food access/referrals or food education, remains sparse [13,14]. To date, there is no standardized design for programs addressing food insecurity. Although programs inevitably vary based on local needs and available resources, establishing a standard framework could be beneficial for future initiatives.

The “Geaux Get Healthy Clinical Program at Our Lady of the Lake” Hospital (GGHOLOL) is a clinic-based community program that combines both education and access to resources for participants while leveraging community partnerships and a clinical setting as an intervention to improve food insecurity. It was created in 2020 through community partnerships to address food insecurity in Baton Rouge, Louisiana. To examine the effectiveness of the GGHOLOL intervention, the FISH (Food Insecurity and its Sequalae on Health) prospective cohort study was conducted.

## 2. Materials and Methods

### 2.1. Description of the Intervention

GGHOLOL aims to provide both education and access to resources to improve food insecurity in the local community. Residents are screened for food insecurity in both clinic and community settings using the USDA Adult Food Security Module 6-question Screener [15]. This shorter version of the 10-question module is used for screening. These modules categorize food security levels as high, marginal, low, and very low based on responses in relation to food availability, concerns about running out of food and skipping meals due to financial constraints. This survey can be conducted over a period of 12 months or 30 days. Those with a score of at least 2 are eligible for participation in GGHOLOL. To screen clinic patients, clinical staff administer the screening questionnaire during the rooming process, and they notify providers of eligible patients. The provider informs the patient about GGHOLOL and refers interested patients to the program. To screen the community at large, we distributed flyers with a QR code link to the online screener. Once completed, the survey was routed to the community health worker, who contacted eligible participants to provide more information and, upon interest, guidance for enrollment.

GGHOLOL includes in-person, hands-on cooking classes, a nutrition class, and a grocery store tour using the American Heart Association Healthy for Life Curriculum^®^. The classes occur within the community classroom at a clinic in North Baton Rouge on weekdays (usually Tuesdays and Thursdays) from 4:30 to 6 p.m. Each participant had a cooking station setup with ample supplies including, but not limited to, a burner, pan, cutting board, bowl, chef’s knife, etc. Recipes and associated health/nutrition educational materials were also at each station. A teacher trained by an American Heart Association trainer through a train-the-trainer model led the lesson. The classes included both group education and hands-on guidance with floating community volunteers as the teacher ambulated around the room with intermittent pauses to demonstrate the technique. Participants were encouraged to bring one family member or friend for support. Educational topics included sodium and sugar restrictions in relation to hypertension and diabetes, portion control, food safety, chopping skills, eating seasonal produce, etc. Although the nutrition, cooking skills, and safety educational content remained the same, GGHOLOL, in partnership with AHA, has enhanced the Healthy for Life curriculum to be more health-literate and culturally relevant based on feedback from GGHOLOL participants and team members. The cooking class and nutrition class took place within the clinic, and the grocery store tour occurred within a Dollar General outfitted with fresh produce adjacent to the clinic.

Safety was of utmost importance during these classes, with precautions outlined within the American Heart Association training and additional safety precautions provided by the OLOL hospital system. Food safety precautions included purchasing foods within packaging from local grocery stores, washing produce, careful handling of raw meat, vigorous cleaning of workspaces and fresh towels, appropriate refrigeration with daily temperature checks, and requiring all participants to wash their hands prior to class. Other safety considerations included having adequate oversight of participants as they worked with sharp knives and hot burners, security was always present during classes, including first aid kits, and as a last resort, an Emergency Room connected to the clinic. These measures were overseen by the OLOL Office of Risk Management.

As a part of the program, participants received educational materials, including information about community partners and ongoing community health events, one free box of fresh produce and frozen chicken from TopBox^®^ (either delivered to the clinic or their home), one $10 gift card to Dollar General, and a set of measuring cups and spoons. Participants also received bus passes for transportation and any other assistance with other social determinants of health that the community health workers (CHW) could provide if needed. While enrolled, the CHW provides ongoing support to participants by reaching out to remind them of upcoming classes, answer questions, and address any barriers to attendance. The CHW also provides participants with ongoing access to other resources and events addressing food insecurity in the community through GGHOLOL partnerships with the local government, a large hospital system, national organizations, and local community-based organizations.

Over two years, adaptable elements of the program have changed. The initial program in 2020 lasted 12 weeks, including 5 cooking classes and 1 combination nutrition class with a grocery store tour. However, based on participant feedback, the GGHOLOL leadership shortened the program to 8 weeks, comprised 3 cooking classes and 1 combination nutrition class and grocery store tour in 2021 and 2022. Despite these small changes, the predefined core components of the program remained the same, with the same average number of educational experiences and average time of completion of the program. All participants were considered to have completed the program if they attended at least 3 educational experiences. These participants were all invited to participate in a celebratory event with food, fellowship, certificates, and local guest speakers to commend them on their efforts.

The Geaux Get Healthy Clinical Program at Our Lady of the Lake is unique as it is founded on a partnership between a hospital system and community partners. It is a joint venture of Our Lady of the Lake Hospital and LSU Health Baton Rouge and is one of many partners in the larger Geaux Get Healthy Coalition (GGH) in Baton Rouge, Louisiana. The Geaux Get Healthy Coalition is a collective impact of organizations that is backboned by the Mayor’s Healthy City Initiative (HealthyBR) and was founded in 2019 through collaborative funding from the Humana Foundation and the Blue Cross Blue Shield of Louisiana Foundation. The GGHOLOL program joined in 2020. GGH has had varying partners over the years, but some consistent GGH-partnered organizations include the American Heart Association, TopBox Foods, and Baton Roots. These other organizations have supported GGHOLOL in many ways. The American Heart Association has provided training for GGHOLOL class leaders through the Healthy for Life curriculum and has assisted with supplies management and providing additional personnel when needed. Topbox delivers a free box of fresh produce and frozen chicken to all GGHOLOL participants, demonstrating the ease and low cost of healthy, locally sourced food delivery. Baton Roots provides opportunities for gardening education and the harvest of fresh produce for free. These organizations have been a constant and viable resource for GGHOLOL participants to further combat food insecurity during and after the completion of the educational program. In addition, GGHOLOL consistently notifies current participants about community events led by our partners, such as community cooking demonstrations, food shares, and gardening education. These events have allowed participants continued access to resources available in their community to encourage sustainable, positive behavior change. HealthyBR was able to provide initial funding for GGHOLOL in 2020 and 2021. Healthy Blue provided funding for GGHOLOL in 2022.

Creating a community program within a clinical setting can be challenging, even with the support of strong community partners. Fortunately, after multiple meetings with stakeholders at Our Lady of the Lake Regional Medical Center (C-suite, human resources, clinic leadership, legal department), we received their unwavering support for GGHOLOL. This was due in large part to the program’s alignment with the hospital’s mission. As with many large systems, internal champions are vital to success, and the clinical department at the OLOL North clinic and the Community Advocacy Committee were those champions for the GGHOLOL program. The program has since formed a strong relationship with OLOL Marketing and the Social Work and Community Health Worker teams. With this support over the years, the GGHOLOL team has grown from one program director, one health specialist (highly skilled community health worker), and a PRN registered dietician to a program director, program manager, scheduled registered dietician, and two community health workers (CHW). This support has been vital for keeping up with every aspect of the program, including CHW appointment schedules, IT and electronic health record troubleshooting, invoicing, data reporting, promotion, refrigerator temperature monitoring, data entry, health fair attendance, ordering groceries, running classes, weekly meetings, etc. The program has also grown to incorporate a rotation for Franciscan Missionaries of Our Lady University-registered dietician interns to learn about community-based nutrition in a unique setting. Similarly, LSU Internal Medicine residents volunteer with the program regularly to gain experience with nutrition education and addressing food insecurity through community outreach. Lastly, it was a challenge to stand up for the program during the height of the COVID-19 pandemic, requiring additional staff and participant safety procedures. This included participant temperature checks, a limited number of guests, social distancing with the grouping of households, masking, strict hand hygiene, and enhanced cleaning protocols. Fortunately, class schedules were not impacted by the COVID-19 pandemic since the classes started in July 2020 once appropriate precautions were in place. The clinical setting was viewed as a “safe” environment for most participants and provided a sense of community during the pandemic. Moreover, participants had fewer conflicting obligations during this time.

### 2.2. Study Procedures

The FISH study is a prospective cohort that studies the effects of the GGHOLOL program. The design is a one-group, pre-post evaluation over two years. FISH eligibility included English-speaking GGHOLOL participants aged 18–65 years without audiovisual deficits. As GGHOLOL participants, all study participants were food insecure as they had all scored at least a 2 on the USDA U.S. Adult 6-question Screening Module to qualify for the program [15]. The primary outcome was an improvement in food insecurity. Secondary outcomes included improvements in health behaviors and depression scores.

### 2.3. Measures

Food insecurity was further evaluated using the more specific USDA U.S. Adult 10-question Food Security Module [15]. A standardized amount of improvement in food insecurity has not been defined in the literature, but since the start of this study, one study quantified improvement as a score reduction by 1 point [16] and another by 2 points [17]. For our purposes, we considered at least a 1-point reduction in the food security score as an improvement but quantified this on a continuous scale instead of a yes/no binary assessment. We assessed improvements in food-related health behaviors using a survey (Appendix A) adapted from Lavelle et al. [18]. In addition to the questions about motivation to prepare healthy meals and cooking/nutrition questions from the validated survey, we asked a few additional questions about food group serving intake. These new questions were evaluated by multiple authors to determine good content validity and proved to have good internal consistency with a mean Cronbach α = 0.84. Lastly, as the gold standard tool for screening depression within a primary care setting, we utilized the Patient Health Questionnaire (PHQ-9) to assess depression [19]. We completed all survey assessments at the initial appointment with the community health worker, again with the community health worker 8–12 weeks later at the completion of the program, and for a third and final time 6 months after enrollment.

### 2.4. Data Analysis

SPSS Statistics (Version 27, IBM Corporation, Armonk, NY, USA) was used for statistical analysis. Descriptive statistics included computing frequencies and percentages for categorical variables and means and standard deviations for continuous variables. The Friedman test was used to analyze changes in response over the three timepoints. The Wilcoxon Signed Rank Test was used to analyze change in response over two timepoints. These data were collected from June 2020 to June 2022 and analyzed in August 2022.

## 3. Results

From June 2020–June 2022, 845 individuals were screened for food insecurity through GGHOLOL, with 628 meeting eligibility for enrollment. Approximately 62% (*n* = 388) of eligible food-insecure individuals were enrolled in GGHOLOL. Subsequently, 237 of the enrolled participants attended at least one class, and 149 participants completed the program by attending at least three classes (38% of eligible food-insecure people). A subset of 120 participants enrolled in GGHOLOL opted to enroll in the FISH research study, with only 57 completing GGHOLOL (Figure 1).

Table 1 shows the baseline demographics of all 120 participants who enrolled in FISH from June 2020 to June 2022. Most participants were African American and female with a mean age of 46. Most participants were referred through the clinic, and the clinic zip code, 70805, was the most frequent zip code of residence for participants (30%). In total, 87% of participants had at least a high school education or GED; the majority had less than USD 25,000 in household income; most had no children living in the household; and 53% were not enrolled in WIC/SNAP. In total, 66% of enrollees had diabetes or pre-diabetes, and 56% had hypertension. Upon enrollment, the mean baseline USDA 10-question Food Security Module score was 5.68 (very low food security), the mean BMI was 34.85 (class 1 obesity), the mean waist circumference was 42.23 (with increased abdominal adiposity for both men and women), and the mean PHQ-9 score was 6.97 (mild depression). Most participants had standard home appliances.

### 3.1. Food Insecurity

Of the 57 participants who completed GGHOLOL, they averaged a score of 6.00 on enrollment, equating to very low food security (raw score 6–10). The mean of the food security score improved to 2.67 upon the completion of the Geaux Get Healthy Clinical Program, which rounded up to 3, correlating with low food security (raw score 3–5). The mean food security score further improved at 6 months to 2.24, which correlated most closely with marginal food security (raw score 1–2). All these measures were statistically significant (*p* = 0.001). This is demonstrated in Figure 2.

### 3.2. Food Serving Consumption

Participants consumed a mean of 1.47 servings of fruit daily upon enrollment in the study and 2.15 servings per day upon completion, which was sustained at 2.18 servings at 6 months. This was statistically significant across all three timepoints (*p* = 0.006). This information included 55 participant answers upon enrollment, 46 at program completion, and 33 at 6 months.

Vegetable serving information included 56 participant answers upon enrollment, 46 at program completion, and 32 at 6 months. Participants consumed a mean of 1.75 servings of vegetables daily upon enrollment, which improved to 2.87 servings per day upon the completion of the program; however, this metric decreased to 2.63 servings of vegetables daily at 6 months. This increase in the mean of more than one full serving of vegetables from enrollment to program completion was statistically significant (*p* = 0.004) and also significant over all three timepoints (*p* < 0.001).

Figure 3 shows the improvement in fruit and vegetable consumption. The grain consumption question only included 27 answers upon enrollment since this question was added to the survey later, and these results were not significant, with 1.41 servings at enrollment, 1.86 servings at program completion, and 1.53 servings at 6-month follow-up.

### 3.3. Other Nutrition Behaviors

On a range of “not confident at all” [1] to “very confident” [5], participants answered, “how confident are you in your ability to follow a recipe?”. The mean was 4.27 at enrollment, improved to 4.67 at program completion, and further improved to 4.74 at 6 months (*p* = 0.03). Using this same scale, participants also answered, “how confident are you in your ability to prepare a healthy meal?” The mean was 4.13 at enrollment, 4.67 at program completion, and 4.65 at 6 months (*p* = 0.012). This is demonstrated in Figure 4.

Lastly, on a range of “never do this” [1] to “always do this” [5], participants answered the question, “when purchasing food, I read the food label and check nutritional values”. This question was added later, so it only included 18 participants’ answers after 6 months. The mean was 2.50 on enrollment, 4.44 upon program completion, and 4.56 at 6 months. This behavior significantly improved (*p* < 0.001). This is demonstrated in Figure 5.

### 3.4. Depression

The mean PHQ-9 score on enrollment was 6.77, which correlates with mild depression (PHQ-9 scores 5–9). As shown in Figure 6, the mean PHQ-9 score improved to 3.33 upon program completion and remained steady at 3.56 during the 6-months follow-up. Scores 0–4 were considered normal scores without signs of depression.

## 4. Discussion

In this prospective cohort of adults with food insecurity, participation in the GGHOLOL program resulted in a statistically significant improvement in food security scores at program completion, with sustained improvement at 6 months. Although we know that SNAP, which benefits and directly provides food, is associated with improvements in food insecurity [11,12,13,14], there is limited knowledge in the literature to date that describes an improvement in food insecurity utilizing a combination intervention utilizing both hands-on cooking and nutrition education with ongoing access to community resources through a community health worker. There is one recent study that suggests hands-on nutrition and cooking education in combination with providing fresh produce, which can increase the amount of fresh produce purchased but not an improvement in food security scores [20]. GGHOLOL provides both education and food access but also leverages partnerships with the local government, national organizations, and local community-based organizations to provide a multitude of other resources and ongoing support. The results of this study are a result of the programming of GGHOLOL in conjunction with the ongoing support of a larger GGH coalition. These collaborative partnerships, along with their establishment within a hospital system promoting food insecurity as a risk factor for poor health, likely contributed to the improvement in food security scores seen. Furthermore, programs such as GGHOLOL can serve as an additional resource for people who are food insecure, especially for those not receiving SNAP benefits, like 53% of people in this study.

We observed statistically significant improvement in fruit and vegetable intake, which has been seen in other community-based cooking skill interventions [21]. Vegetable intake increased to meet and surpass the USDA (United States Department of Agriculture)-recommended amount of 2.5 servings daily, and fruit intake increased above the recommended 2 servings daily [22]. However, the improvement in fruit intake may not be clinically significant as this was not an increase by one full serving. This increased intake of healthful foods is likely a reflection of an awareness of local resources, an improved knowledge of available foods improved knowledge of associated health benefits, access to recipes, and increased awareness and excitement about healthy eating overall. The improved adherence of our participants to USDA dietary guideline recommendations for fruit and vegetable intake could have positive implications on their health as these guidelines are created from robust evidence to promote well-being through healthy nutrition recommendations for all Americans. Lastly, the whole grain intake was not increased. However, the lack of significance with the increase in the whole grain intake may be impacted by the low number of responses since this question was added later.

Participants also showed improvements in cooking and nutrition behaviors, such as confidence in following a recipe, confidence in preparing a healthy meal, and the frequency of reading food labels. These improvements in cooking confidence and reading food labels suggest that the Healthy for Life curriculum by the American Heart Association is effective in improving cooking and nutrition behaviors when provided with support from a CHW and other partnering organizations. GGHOLOL promotes these behaviors through the enhancement of cooking skills, improved nutrition knowledge, and real-life experiences of preparing healthy recipes and practicing reading food labels with appropriate guidance. Other culinary education programs have also demonstrated an improvement in cooking confidence, particularly in vulnerable populations with low food literacy [21]. People with more confidence in following a recipe and preparing a healthy meal may be more able and likely to prepare healthy meals at home, especially when we see a concomitant increase in fruit and vegetable consumption in this population [21]. Furthermore, it is reported that those with more confidence in cooking behaviors are more likely to have a normal BMI [23], and reading a food label is associated with a lower prevalence of metabolic syndrome [24].

Lastly, an unexpected improvement in GGHOLOL intervention included a statistically significant improvement in depression with an improvement in the mean PHQ-9 score. Many of our participants described the program as uplifting as it provided a sense of community and a way to take control of their health. Participants also voiced that they felt valued and supported by GGHOLOL. Since the start of this study, Rees et al. published the impacts of a community-based cooking intervention on mental health with an overall improvement in mental health scores [23]. This phenomenon has been seen in other studies, as well with other community-based culinary classes promoting a communal spirit along with the association of improved dietary intake and improved mental health [23]. This sense of community and self-efficacy seen within and amongst our participants, which is to be described further in a qualitative evaluation of GGHOLOL, likely contributed to the improvements in the mean PHQ-9 score and corroborated with the findings in these other studies.

Through nutrition education and access to resources, the Geaux Get Healthy Clinical Program at Our Lady of the Lake has improved mean food security scores in a food-insecure population with sustainable improvement. This program also improved healthy eating, cooking, and shopping behaviors. Furthermore, mean depression scores (PHQ-9) improved upon program completion with sustainable improvement at 6 months. These improvements in food insecurity, healthy eating behaviors, and mood are likely to positively impact the overall health and well-being of participants.

### 4.1. Limitations

There are many limitations to this study, including the small sample size, high attrition rates, and lack of randomization. We felt that it would be unethical to not offer a program addressing food insecurity to food-insecure individuals with limited resources in the community. Therefore, we offered GGHOLOL and FISH research participation, knowing that this population typically has high “no show” rates. Those enrolled in the FISH research study had self-selection bias. These GGHOLOL participants are often those most excited about participating in the cooking classes. Furthermore, those who continued through each evaluation despite the high dropout rate in this study brought about attrition bias. Although there is no consensus on an acceptable dropout rate in studies involving vulnerable populations, the dropout rate of 52% is unsatisfactory. The reasons for this high dropout rate are currently under investigation.

Other limitations of this study include minor changes within the Geaux Get Healthy Clinical Program over 2 years due to the nature of the program. Although based in a clinic, this program functions as a community program with community partners and grant funding, leading to changes as a reflection of the evolution of community partnerships, changes based on participant feedback, and changes in grant funding. Some of these changes included transitioning the length of the program from 12 weeks to 8 weeks and decreasing 5 cooking classes to 3 cooking classes. However, the pre-identified core components of the program remained consistent. Further, we were unable to control changes brought on by the COVID-19 pandemic, including changes in employment, unemployment benefits, inflation, government payments, and fluctuations in food stamp benefits. Lastly, we could not determine the long-term effects of such a program on chronic disease prevention and management at this time.

### 4.2. Implications for Research and Practice

There are many opportunities for ongoing research in this field, as evidenced by multiple “Food Is Medicine” initiatives throughout the country focusing on health policy, health equity, and research [25,26,27]. Specifically, the American Heart Association Food Is Medicine Initiative is looking to support pragmatic research to guide program design using lived experiences that promote participant engagement and improvement in healthy eating behaviors [25]. With the results thus far obtained, we hope to use the challenges and successes of FISH and GGHOLOL to promote further pragmatic research on “Food is Medicine” and food insecurity. In the future, we hope to describe GGHOLOL’s impact on markers of metabolic health, including waist circumference, weight, lipid panels, and hemoglobin A1Cs. The long-term follow-up of these participants may allow us to look at patient-centered health outcomes such as major adverse cardiovascular events and mortality. We also plan to provide an overall cost analysis. Further, we currently have two qualitative studies looking at reasons why people who are food insecure are not always enrolled in the Supplemental Nutrition Assistance Program (SNAP) with benefits and barriers to follow-up in a food insecure population. There is a large need for research on attrition rates in populations with social determinants of health needs, as this could be beneficial for all future programming and studies in vulnerable populations on any topic.

## 5. Conclusions

The Geaux Get Healthy Clinical Program at Our Lady of the Lake improved food security scores, healthy eating behaviors, and mood in a vulnerable, food-insecure population in Baton Rouge, Louisiana. GGHOLOL may serve as a model for other clinic-based community programs addressing food insecurity in the future. This program is unique due to the comprehensive education, access to resources, and community partnerships rooted within a hospital system. The success of GGHOLOL may encourage insurance companies, large payers, and healthcare systems to incorporate such models to improve downstream food insecurity and future health in their patient population.

## Figures and Tables

**Figure 1 nutrients-15-04316-f001:**
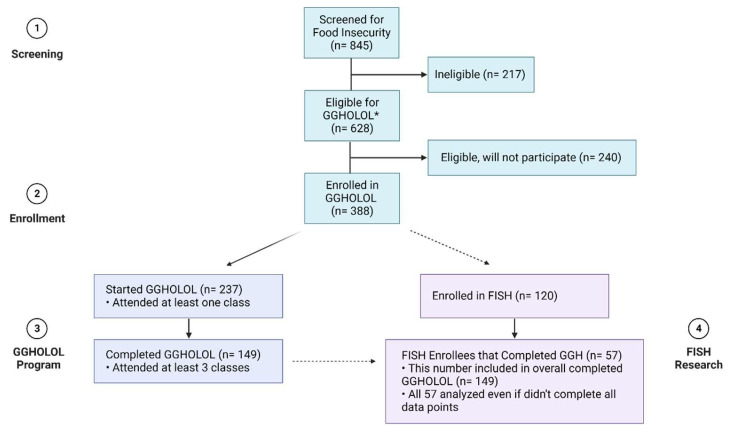
Enrollment flowchart. * scored at least a 2 on the USDA U.S. Adult 6-question Screening Module.

**Figure 2 nutrients-15-04316-f002:**
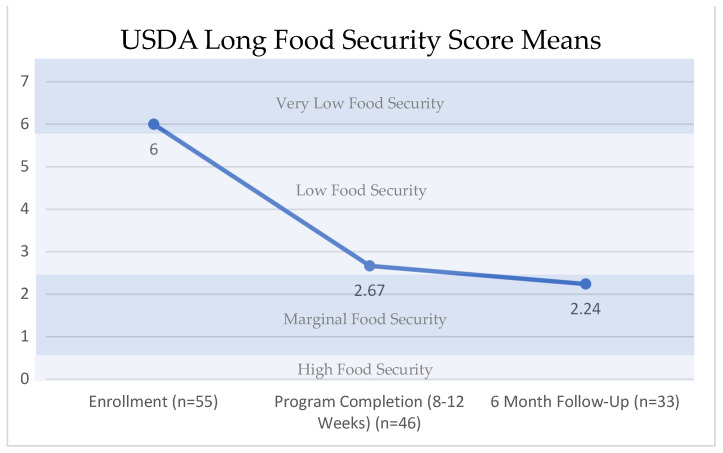
USDA U.S. Adult Food Security Survey Module Means. Friedman’s Two-Way Analysis of Variance by Ranks Summary (*p* = 0.001).

**Figure 3 nutrients-15-04316-f003:**
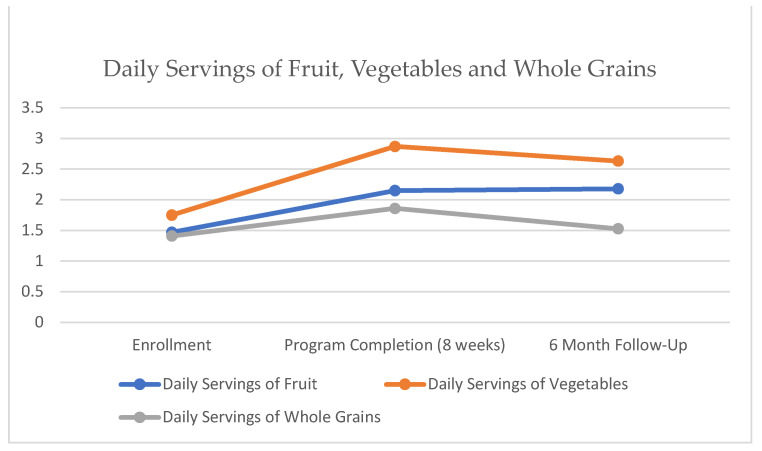
Daily servings of fruit, vegetables and whole grains. Friedman’s Two-Way Analysis of Variance by Ranks Summary (fruit *p* = 0.006, vegetables *p* < 0.001, whole grains *p* = 0.24).

**Figure 4 nutrients-15-04316-f004:**
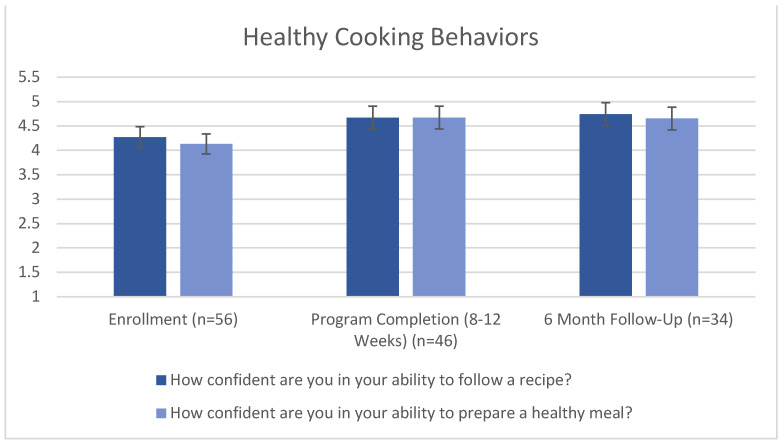
Healthy cooking behaviors. Friedman’s Two-Way Analysis of Variance by Ranks Summary (*p* = 0.03) for recipe. Friedman’s Two-Way Analysis of Variance by Ranks Summary (*p* = 0.012) for healthy meal. Error bars indicate 95% confidence interval.

**Figure 5 nutrients-15-04316-f005:**
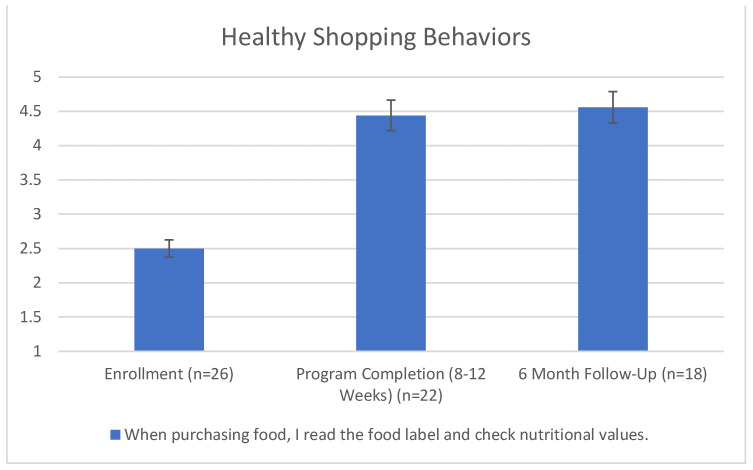
Healthy shopping behaviors. Friedman’s Two-Way Analysis of Variance by Ranks Summary (*p* < 0.001). Error bars indicate 95% confidence interval.

**Figure 6 nutrients-15-04316-f006:**
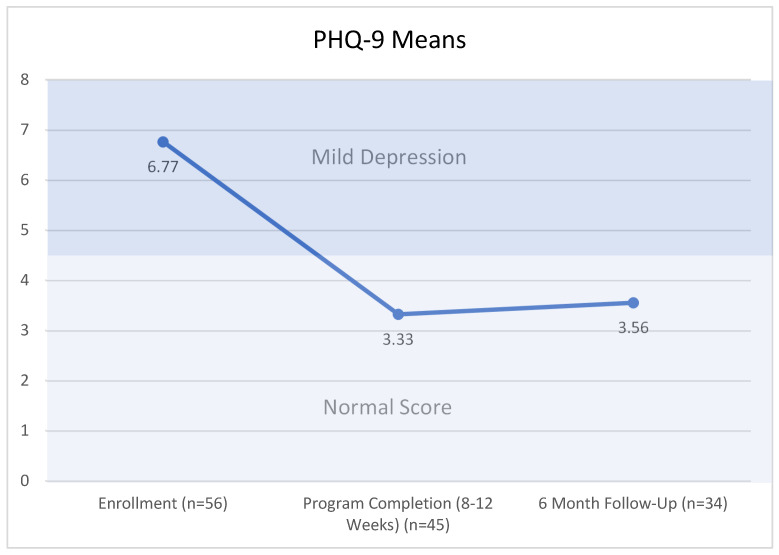
PHQ-9 Means. Friedman’s Two-Way Analysis of Variance by Ranks Summary (*p* = 0.034). Related-Samples Wilcoxon Signed Rank Test summary enrollment vs. program completion (*p* = 0.002). Related-Samples Wilcoxon Signed Rank Test Summary Program completion vs. 6 month follow-up (*p* = 0.8). Related-Samples Wilcoxon Signed Rank Test summary enrollment vs. 6 month follow-up (*p* = 0.01).

**Table 1 nutrients-15-04316-t001:** Baseline characteristics (*n* = 120).

Baseline Characteristics (*n* = 120)	
	*n* or Mean (% or SD)
Race	
African American	101 (84.2%)
White	5 (4.2%)
Hispanic	4 (3.3%)
Other	10 (8.3%)
Gender	
Female	95 (79.2%)
Male	23 (19.2%)
Other	2 (1.6%)
Age	
18–24	10 (8.4%)
25–34	11 (9.2%)
35–44	33 (27.7%)
45–54	23 (19.3%)
55–64	40 (33.6%)
65	2 (1.7%)
Referral location	
Clinic	98 (81.7%)
Community	22 (18.3%)
Education	
Master’s Degree or Higher	3 (2.5%)
Completed College	7 (5.9%)
Some College or Vocational School	39 (33.1%)
High School or GED	54 (45.8%)
Less than High School	15 (12.7%)
Annual Household Income	
Less than USD 25,000	88 (74.6%)
USD 25,000–USD 34,999	19 (16.1%)
USD 35,000–USD 49,999	6 (5.1%)
USD 50,000–USD 74,999	5 (4.2%)
More than USD 75,000	0 (0%)
Children in Household	
0	68 (59.1%)
1	22 (19.1%)
2	10 (8.7%)
3	9 (7.8%)
4+	6 (5.2%)
Receive WIC/SNAP	
Yes	55 (46.6%)
No	63 (53.4%)
Metabolic Conditions	
Pre-diabetes	35 (29.2%)
Diabetes	44 (36.7%)
Hypertension	67 (55.8%)
Metabolic Measurements	
Average BMI (kg/m^2^)	34.85 (9.23)
Average Waist Circumference (in)	42.23 (7.94)
Mean PHQ-9	6.97 (6.34)
Home appliances	
Stove/hotplate	113 (94.2%)
Oven	114 (95.0%)
Microwave	113 (94.2%)
Pan	113 (94.2%)
Refrigerator	118 (98.3%)

## Data Availability

The data presented in this study are available on request from the corresponding author. The data are not publicly available due to storage in an institutional, HIPAA-compliant database with access to identifying information.

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
