# Peer review of "The Design and Impact of a Clinic-Based Community Program on Food Insecurity, Healthy Eating Behaviors, and Mood"

_nutrients, 2023, doi:10.3390/nu15204316_

Round 1

Reviewer 1 Report

In this paper Impact of a Clinic-based Community Program on Food Insecurity, the Geaux Get Healthy Clinical Program at Our Lady of the Lake (GGHOLOL) is a clinic-based community program that leverages community partnerships and a clinical setting to provide education and access to resources to individuals with food insecurity. This prospective study examines the impact of GGHOLOL on food insecurity as a pre-post survey evaluation over a two-year period. The article is in line with the readers interest of Nutrients. However, there are still some obvious shortcomings.

Comments:

Q1. The title does not align with the content of the article in my opinion. Upon reading it, I perceive a focus on the adverse effects of various food-related risk factors that currently impact us.

Q2. How could authors define the 57 participants with food insecurity?

Q3. The number of references is insufficient and it is advisable to augment them to 40-50.

Q4. The criteria for assessing the safety of food should be explained clearly.

Q5. Whenever feasible, it is advisable to complement the mechanisms or potential causes of the health effects associated with each intervention.

Q6. Please check the presentation of (p=0.034). Generally, the format should be p<0.05, 0.01, 0.001, etc.

Reviewer 2 Report

Overall, the paper is acceptable. It is addressing an important issue in well, planned and methodologically sound manner. It presents important and encouraging results.  It has several significant limitations, however. Due to the various points, where attrition eats away at the sample size, the researchers are left with a very small number of participants, whom, as they acknowledge, are likely self selected to be those most likely to implement the educational program at home and gain the benefits.  I was also somewhat bothered by several references to additional supportive programs from community partners, and/or the healthcare system involved, but these additional programmatic factors or supports were not detailed, and it's not clear if the positive results from this study were gained independently, or only in conjunction with the entire support net. Finally, no mention is made of Covid and how the program worked with in that timeframe.   Surely this had an impact on participation?  What about changes to participants diets due to either negative impacts from the pandemic or from increased financial support during that time for many programs impacting this target population?  I found this very odd.  Overall, I would have been more comfortable with language that was more limiting in the description of the results.  I do have to say, I appreciate that the authors explicitly comment on the concerning drop out rate at all stages of the process.  

Round 2

Reviewer 1 Report

Thanks for these modifications and explanations, I have no other questions.